# The Hippo Pathway Effector YAP1 Regulates Intestinal Epithelial Cell Differentiation

**DOI:** 10.3390/cells9081895

**Published:** 2020-08-13

**Authors:** Sepideh Fallah, Jean-François Beaulieu

**Affiliations:** 1Laboratory of Intestinal Physiopathology, Department of Immunology and Cell Biology, Faculty of Medicine and Health Sciences, Université de Sherbrooke, Sherbrooke, QC J1H 5N4, Canada; sepideh.fallah@usherbrooke.ca; 2Centre de Recherche du Centre Hospitalier Universitaire de Sherbrooke, Sherbrooke, QC J1H 5N4, Canada

**Keywords:** Hippo pathway, YAP1, TAZ, intestinal cell, differentiation, CDX2, stem cell, goblet cell, absorptive cells

## Abstract

The human intestine is covered by epithelium, which is continuously replaced by new cells provided by stem cells located at the bottom of the glands. The maintenance of intestinal stem cells is supported by a niche which is composed of several signaling proteins including the Hippo pathway effectors YAP1/TAZ. The role of YAP1/TAZ in cell proliferation and regeneration is well documented but their involvement on the differentiation of intestinal epithelial cells is unclear. In the present study, the role of YAP1/TAZ on the differentiation of intestinal epithelial cells was investigated using the HT29 cell line, the only multipotent intestinal cell line available, with a combination of knockdown approaches. The expression of intestinal differentiation cell markers was tested by qPCR, Western blot, indirect immunofluorescence and electron microscopy analyses. The results show that TAZ is not expressed while the abolition of YAP1 expression led to a sharp increase in goblet and absorptive cell differentiation and reduction of some stem cell markers. Further studies using double knockdown experiments revealed that most of these effects resulting from YAP1 abolition are mediated by CDX2, a key intestinal cell transcription factor. In conclusion, our results indicate that YAP1/TAZ negatively regulate the differentiation of intestinal epithelial cells through the inhibition of CDX2 expression.

## 1. Introduction

The luminal surface of the mammalian intestine is covered by single layer of epithelial cells, which not only digest and absorb nutrients but also form a protective layer against pathogens [1]. The epithelium of the intestine has one of the highest turnover rates in the body, being continuously replaced by new cells provided by LGR5 positive crypt base columnar (CBC) stem cells [2,3,4]. CBC stem cells symmetrically and asymmetrically divide to generate new stem cells and transit amplifying (TA), or progenitor, cells [5,6]. Absorptive progenitors proliferate and give rise to the absorptive cells (enterocytes), which compose more than 80% of epithelial cells, while secretory progenitors have limited proliferation and differentiate into Paneth, goblet and enteroendocrine cells [7,8]. Enterocytes are simple columnar epithelial cells characterized by the brush border on their apical surface to increase the surface for digestion and absorption of nutrients. Sucrase-isomaltase (SI), which cleaves disaccharides and oligosaccharides to monosaccharides [9], and dipeptidyl peptidase IV (DPPIV/CD26),which cleaves dipeptides from the *N*-terminus of polypeptides [10], are two prominent markers of absorptive cells [1,11]. The main secretory cell type, goblet cells, produces highly glycosylated gel-forming mucins, MUC2 and trefoil factor 3 (TFF3), which limits bacterial access and stabilizes the mucus layer [12], while Paneth cells secret antimicrobial peptides including alpha defensins (DEFAs) and enteroendocrine cells release a variety of hormones such as chromogranin A (CHGA) into the blood stream [13,14].

Although not yet completely understood, there are several factors known to be involved in the regulation of intestinal epithelial cell self-renewal and the differentiation sequence such as soluble and insoluble niche elements, hormones and growth factors [15]. The Wnt/β-catenin and Notch signaling pathways are two important ones [3,16,17,18]. The Wnt/β-catenin pathway, which has its highest activity at the base of the crypt, not only regulates stemness but also plays an important role in Paneth cell differentiation and homeostasis of the intestine [16,19]. The Notch signaling pathway, which is activated by the binding of JAG and DLL ligands to the Notch receptor, promotes absorptive cell differentiation through the stimulation of HES1 expression and the inhibition of ATOH1 transcription factors. ATOH1 expression is required for secretory cell differentiation while inactivation of the Notch signaling leads to the higher expression of ATOH1 and secretory cell hyperplasia [16,18]. There are many other transcription factors that have been characterized for their effects on intestinal epithelial cell proliferation and differentiation [20]. Among them are CDX2, HNF-1α and GATA-4, which cooperate to modulate intestinal cell differentiation and proliferation [21,22]. CDX2 inhibits colorectal cancer cell proliferation and its low expression in colorectal tumors is linked to a less favorable prognosis [23,24]. On the other hand, knockout of HNF-1α in the mouse results in increased crypt proliferation, a decreased number of enteroendocrine cells, altered Paneth cell maturation and finally a disturbance of intestinal epithelial cells during adulthood [25]. Another of these key transcription factors is KLF4, a zinc finger transcription factor which appears to be required for terminal differentiation of goblet cells [26]. Its role in the reprogramming of embryonic stem cells, homeostasis of the intestine and suppression of colorectal cancer has been reported previously [27].

More recently, an important role of the Hippo pathway effector Yes associated protein 1 (YAP1) and its paralog transcriptional co-activator with PDZ-binding motif (TAZ) has been pointed out on the regulation of intestinal cell proliferation, differentiation and tumorigenesis [28]. The Hippo pathway, which is composed of upstream signals, a kinase core and downstream target genes, restricts overgrowth of the tissue through the phosphorylation and inactivation of the YAP1/TAZ coactivators. Upstream signals such as cell contact, mechanical signals including stiffness and extracellular matrix composition, hormonal signals and growth factors regulate the Hippo pathway and YAP1/TAZ activity [29]. Initiation of kinase core activity is established by MST1/2 kinase activation, which phosphorylates and activates the LATS1/2 kinases by helping the scaffolding protein WW45. Phosphorylated LATS1/2 with the regulatory protein MOB1 phosphorylates and inactivates the YAP1/TAZ co-activators [29]. Phosphorylated YAP1/TAZ is restrained to the cytoplasm and is recognized by 14.3.3 or SCF^beta-TrCP^ E3 ubiquitin ligase for ubiquitination and proteasomal degradation [30,31]. Absence of Hippo pathway activity leads to the entrance of active YAP1/TAZ into the nucleus and formation of the YAP1/TAZ-TEAD complex, which finally activates transcription of the genes involved in cell growth and survival. CYR61 and CTGF are among the well-known YAP1/TAZ downstream target genes [32,33]. Several previous studies have shown YAP1/TAZ to have a role in cell proliferation, regeneration and reprogramming [34,35,36]. Panciera and colleagues observed that induced YAP1 expression converts the differentiated mammary gland, pancreatic exocrine and neuron cells into stem/progenitor cells [35]. On the other hand, knockdown of YAP1/TAZ expression leads to the differentiation of insulin-producing β cells [37]. During colonic epithelium repair in the intestine, fetal-like tissue is formed through YAP1/TAZ activation, which is induced by the remodeling of the extracellular matrix (ECM). The high level of collagen type 1 in the ECM of repairing epithelium results in increased activity of FAK/Src signaling, which controls YAP1/TAZ activity. Thus, dedifferentiation and modification of the committed cells into stem/progenitor cells has been observed in ectopic YAP1/TAZ expression [34]. Recent findings have also shown the interaction of other signaling pathways including Wnt, Notch and epidermal growth factor (EGF) with the Hippo pathway and YAP1/TAZ co-activators [32,38]. In this context, it is interesting to note that during intestinal regeneration YAP1 participates in intestinal stem cell maintenance through the temporary suppression of Wnt signaling [36]. While Wnt signaling activity and Paneth cell differentiation are both restricted by active YAP, its removal leads to a higher expression of lysozyme, a Paneth cell marker [36,39]. A potential dual role of YAP1/TAZ has been reported in the intestinal epithelium. Imajo et al. [40] showed that YAP1/TAZ has a stimulatory effect on stem/progenitor cell proliferation when combined with TEAD while it can promote goblet cell differentiation in complex with KLF4. Thus, the exact role of YAP1/TAZ in the regulation of intestinal cell differentiation remains ambiguous.

In this study, we used the multipotent intestinal cell line HT29 to further investigate the involvement of the Hippo pathway on the differentiation of human absorptive and secretory intestinal lineages. Indeed, HT29 cells exhibit the unique ability to grow under normal conditions as a multilayer of undifferentiated cells which, similar to other cancer cell lines, contains stem-like cells [41] that have the potential to differentiate into absorptive and/or goblet cells under specific conditions [42,43,44].

## 2. Materials and Methods

### 2.1. Cell Culture, shRNAs and Lentiviral Infection

The human colorectal cancer cell line HT29, which is available from the American Type Culture Collection (Manassas, VA, USA), and the Caco-2/15 cell line [45] were obtained from A. Quaroni (Cornell University, Ithaca, NY, USA). HT29 and Caco-2 identities were confirmed by short-tandem repeat profiling cell authentication. Although cancerous in nature, HT29 and Caco-2/15 have been widely used as experimental models to study human small intestinal cell differentiation since, when grown under adequate conditions, they can recapitulate goblet (HT29) and absorptive (HT29 and Caco-2) cell morphological and functional differentiation patterns similar to those of the human small intestine. The utility and limitations of these models are well documented [42,43,44,46,47,48]. Cells were cultured in Dulbecco’s modified Eagle’s medium (Life Technologies, Burlington, ON, Canada), supplemented with 10% fetal bovine serum (Wisent, Saint-Jean-Batiste, QC, Canada), 2 mM GlutaMAX (Life Technology) and 10 mM HEPES Wisent). The HIEC cells used herein as normal control cells in some experiments were derived from the normal small intestine and grown as described elsewhere [49]. All cells were maintained in a 5% CO_2_-humidified atmosphere at 37 °C. Regular monitoring was performed to insure the absence of mycoplasma contamination.

For the knockdown of YAP1 and CDX2 expression, lentiviruses were prepared with MISSION shRNA (Sigma-Aldrich, Oakville, ON, Canada) plasmids containing shRNA targeting YAP1 and CDX2 obtained from Addgene (Watertown, MA. USA) and Sigma-Aldrich (Canada). The shYAP1 #1 and shYAP1 #2 sequences were 5′-GCCACCAAGCTAGATAAAGAA-3′ and 5′-CCCAGTTAAATGTTCACCAAT-3′; respectively. The two shRNAs were a gift from William Hahn [50] (Addgene plasmids # 42,540 and 42541). The shCDX2 was the construct TRCN0000013687 targeting the sequence 5′-AGCCCTTGAGTCCGGTGTCTT-3′. A shRNA targeting TAZ (construct TRCN0000019469 targeting the sequence 5′GCGATGAATCAGCCTCTGAAT-3′) was used as a negative control for all experiments with HT29 cells. In some preliminary experiment where Caco-2/15 cells were tested, control shRNAs used were shGFP [51] and shLUC [52].

For HT29, cells at 50% confluency were infected with lentivirus. At 48 h post-infection, stable cell lines were selected by adding 10 μg/mL of puromycin (shYAP1 and shTAZ) and 1 mg/mL of G418 (shCDX2) for 9 days. Cells were then plated and grown to confluence and harvested at Day 0 and 8 days post confluence for HT29shYAP1 and HT29shTAZ and Day 5 post confluence for HT29shYAP1+shCDX2. For Caco-2/15, cells were infected and selected as described above for shYAP1 except that shGFP and shLUC were used as controls since these Caco-2/15 cells express TAZ. The cells were tested at Day 5 post confluence.

### 2.2. Tissues

Normal adult human intestinal tissues (jejunum and ileum) were provided by Transplant Quebec (Québec, Canada) according to a protocol approved by the Institutional Human Subject Review Board of the Centre Hospitalier Universitaire de Sherbrooke.

### 2.3. Antibodies

The primary antibodies used in this study include mouse monoclonal anti-MUC2 (Abcam, ab11197, 1/500 WB and 1/100 IF), rabbit monoclonal anti-TFF3 (ab108599, 1/1500 WB, Abcam, Toronto, ON, Canada), mouse monoclonal anti-SI ([53]; HSI-4/34 or Caco-3/73, 1/100 WB), rabbit polyclonal anti-DPPIV or CD26 (ab129060, 1/2000 WB, Abcam), mouse monoclonal anti-DPPIV (ab3154, 1/100 WB, Abcam), mouse monoclonal anti-DPPIV ([54] DAO 7/219, 1/100 IF, a gift from A. Quaroni), rabbit monoclonal anti-YAP/TAZ (D24E4, 1/1500 WB, 1/50 IF, Cell Signaling, Danvers, MA, USA), mouse monoclonal anti-CDX2-CD88 (MU392A-UC, 1/700, BioGenex, Freemont, CA, USA), mouse monoclonal anti-LGR5 (UMAB212, 1/100 WB and 1/100 IF, Origene, Rockville, MD, USA) and anti-β-actin (MAB1501, 1/20,000, Millipore, Etobicoke, ON, Canada). The secondary antibodies used in this study include AlexaFluor 488 or 594 goat anti-mouse (A11017, A11072, 1/400, and goat anti-rabbit (A11070, A11072, 1/400; Thermo Fisher Scientific, Ottawa, ON, Canada), ECL HRP-linked anti-mouse (NA931 V, 1/4000, GE Healthcare, Mississauga, ON, Canada) and anti-rabbit (NA934 V, 1/4000).

### 2.4. Western Blot Analysis

Cells were harvested for protein extraction using Laemmli 1× buffer. The samples were sonicated and equal amounts of reduced (5% β-mercaptoethanol) protein samples were migrated through 10%, 12% or 15% SDS-PAGE (polyacrylamide gel electrophoresis). Two percent vertical agarose gel electrophoresis was performed for the separation of MUC2 protein which has a molecular weight of about 500 KD. Briefly, agarose powder (Roche, Penzberg, Germany) was dissolved in Nano pure water by heating in a microwave for 90 s. Then 20× gel™ running buffer was added to make a 1× final solution. Melted agarose was poured into a vertical casting unit and a well comb was inserted. After 20 min, the gel was transferred to 4 °C for 30 min. Finally, the samples were migrated for 45 min at 100 volts. For all WB experiments, nitrocellulose membrane (GE Healthcare, Mississauga, ON, Canada) was used for transferring and 5% non-fat milk was used to block nonspecific binding sites. The membranes were incubated overnight with primary antibodies at 4 °C, then rinsed with 0.1% PBS-tween and incubated with horseradish peroxidase-conjugated secondary antibodies. After rinsing the membrane, the enhanced chemiluminiscence (ECL) method was used to detect HRP positive bands according to the manufacturer’s instruction (Millipore, WBKLS0100). Image J [55] (National Istitute of Health, Bethesda, MD, USA) was used for scanning and band quantitation.

### 2.5. RNA Extraction, Reverse Transcriptase and Quantitative RT-PCR

RiboZol (AMRESCO, Solon, OH, USA) was used for cell lysing. RNA extraction, reverse transcription and quantitative polymerase chain reaction (qPCR) assays were performed as described previously for both cells and tissues [56]. SYBR Green Power PCR Master Mix (Bio Basic, Markham, ON, Canada) was used for qRT-PCR. The primers used for qPCR include: YAP1: forward 5′-TGCGTAGCCAGTTACCAAC-3′ and reverse 5′-GGTTCGAGGGACACTGTAGC-3′; WWTR1 (TAZ): forward 5′-TGCTACAGTGTCCCCACAAC-3′ and reverse 5′- GAAACGGGTCTGTTGGGGAT-3′; CTGF: forward 5′-CCTGGTCCAGACCACAGAGT-3′ and reverse 5′-TGGAGATTTTGGGAGTACGG-3′; CYR61: forward 5′-TCCCTGTTTTTGGAATGGAG-3′ and reverse 5′-GAGCACTGGGACCATGAAGT-3′; ATOH1: forward 5′-TGAAGGAGTTGGGAGACCAC-3′ and reverse 5′-TCCGGGGAATGTAGCAAATA-3′; KLF4: forward 5′-GCGGCAAAACCTACACAAAG-3′ and reverse 5′-CCCCGTGTGTTTACGGTAGT-3′; CDX2: forward 5′-GAGTGGTGTACACGGACCAC-3′ and reverse 5′-TTTCCTCTCCTTTGCTCTGC-3′; HNF1α: forward 5′-CCGCAGACTATGCTCATCAC-3′ and reverse 5′-GCTGAGTCTGAGCTCTGGT-3′; MUC2: forward 5′-CATCACATTCATGCCCAATG-3′ and reverse 5′-CAGCTCTCGATGTGGGTGTA-3′; TFF3: forward 5′-CTCCAGCTCTGCTGAGGAGT-3′ and reverse 5′-GAAACACCAAGGCACTCCAG-3′; SI: forward 5′-GAGGACACTGGCTTGGAGAC-3′ and reverse 5′- ATCCAGCGGGTACAGAGATG-3′; DPPIV: forward 5′-AAGTGGCGTGTTCAAGTGTG-3′ and reverse 5′-CAGGGCTTTGGAGATCTGAG-3′; DEFA5: forward 5′-AAGCAGTCTGGGGAAGACAA-3′ and reverse 5′-TGAATCTTGCACTGCTTTGG-3′; CHGA: forward 5′-CGGGAGGACAGCCTTGAG-3′ and reverse 5′-CTGGTGGGCCACTTTCTC-3′; ASCL2: forward 5′-AGCAAGAAGCTGAGCAAGGT-3′ and reverse 5′-GGATGTACTCCACGGCTGAG-3′; OCT: forward 5′-TGCAGAAAGAACTCGAGCAA-3′ and reverse 5′-GTGAAGTGAGGGCTCCCATA-3′; LGR5: forward 5′-TGCTCTTCACCAACTGCATC-3′ and reverse 5′-CTCAGGCTCACCAGATCCTC-3′; PROM1: forward 5′-TTTGGTGCAAATGTGGAAAA-3′ and reverse 5′-TTGAAGCTGTTCTGCAGGTG-3′; CD44: forward 5′-TAAGGACACCCCAAATTCCA-3′ and reverse 5′-CCACATTCTGCAGGTTCCTT-3′; and EPCAM: forward 5′-CACAACGCGTTATCAACTGG-3′ and reverse 5′-CCAGCTTTTAGACCCTGCAT-3′.

Gene expression was calculated according to the Pfaffl equation [57] using RPLPO as a validated normalizer [56] relative to control groups consisting of a pool containing various CRC cell lines [48] or shCtrl as specified in the text.

### 2.6. Indirect Immunofluorescence Staining and Confocal Imaging

Cells were cultured on cover slips and then sub-confluent, and eight-days post-confluent cells were fixed with MeOH for MUC2 or 2% PFA for YAP/TAZ, DPPIV, LGR5 and DEFA5. Five percent fat free milk for MUC2 and DPPIV and 10% goat serum for YAP/TAZ, LGR5 and DEFA5 were utilized as blockers. Primary and secondary antibodies were applied according to the instructions of the manufacturers. Leica DM RXA, Leica DM IRBE and Reichert Polyvar 2 microscopes were used for observing the cells and taking images. The images were acquired using MetaMorph software (Universal Imaging Corporation, West Chester, PA, USA). Stained tissues were also viewed with a confocal microscope Olympus FV1000 SIM equipped with Fluoview for image acquisition. Tissues were embedded in OCT and 3 µm cryosections were stained for YAP1/TAZ and DEFA5 as described above for cells.

### 2.7. Transmission Electron Microscopy (TEM)

The preparation of cell samples for ultrastructural analysis was performed as previously described [58]. Briefly, equal amounts of HT29 expressing shYAP1#1, shYAP1#2 and shTAZ#1 (control) were plated in a 6-well plate. Cells were grown until 8 days post confluence and then washed with PBS three times. The cells were fixed with 1.5% glutaraldehyde for 30 min at room temperature followed by 2.5% glutaraldehyde overnight at 4 °C and post fixed in 1% osmium tetroxide. The samples were dehydrated and embedded in epoxy under vacuum. Sections were prepared on 200 mesh copper grids coated with formvar/carbon film and observed with a Hitachi H-7500 transmission electron microscope at 80 kV.

### 2.8. Statistical Analysis

Data preparation and statistical analysis which included two tailed Student’s *t*-test and ANOVA were performed with Graph Pad Prism 8.3 (Graph Pad Software; San Diego, CA. USA). *p* value < 0.05 was considered significant in all analyses. All experiments were repeated at least three times, independently.

## 3. Results

### 3.1. The Expression of YAP1/TAZ Protein in Human Intestinal Crypt Cells

The expression of YAP1/TAZ protein was detected in the nucleus of some crypt cells located in the stem cell zone. These cells are located between the Paneth cells in which YAP1/TAZ protein was found below detectable levels in their nuclei (Figure 1 and Appendix A) in agreement with previous findings reporting an absence of YAP1 in Paneth cells [36].

### 3.2. HT29 Cells Express Stem Cell Markers and YAP1

HT29 is an undifferentiated colorectal cancer cell line which exhibits some multipotency since these cells express stem cell markers and can differentiate into both absorptive and goblet cells under certain conditions. The expression of stem cell markers *LGR5*, *CD44*, *PROM1*/*CD133*, *EPCAM*/*CD326*, *ASCL2* and *OCT4* as well as the goblet cell marker *MUC2* and absorptive cell marker *SI* was first evaluated in HT29 cells by qPCR analysis. The expression of these transcripts was expressed relative to a pool consisting of a mix of cancer cells including Caco-2/15, HT29, A549 and SKOV3. The results show that five of the six stem cell markers are expressed at high levels in HT29 cells compared with the cancer cell pool. *YAP1* was detected at a comparable level to that of the pool. However, low levels of ASCL2, *TAZ* and *SI* were detected while *MUC2* was expressed at a higher level in HT29 cells compared with the pool (Figure 2A). Western blot analysis confirmed a strong expression of YAP at the protein level in HT29 cells. However, in contrast to Caco-2 cells, another colorectal cancer cell line, TAZ was not detected in HT29. It is noteworthy that YAP1 and TAZ expression in a normal intestinal cell line was also distinct, HIEC expressing only TAZ (Figure 2B). These distinct patterns of YAP1 and TAZ expression were also observed at the transcript level for the three cell lines as well as for the small intestine where relative mRNA levels of YAP1 appeared higher than those of TAZ (Figure 2C). Indirect immunofluorescence analysis demonstrated nuclear expression of the YAP protein in a large proportion of the HT29 cells (Figure 2D). In addition, consistent with LGR5 and MUC2 transcript expression, HT29 were found to constitutively express a subpopulation of stem-like and goblet-like cells (Figure 2D).

### 3.3. Knockdown YAP1 Expression by shRNAs

To determine whether YAP is involved in stemness and/or differentiation, sub-confluent HT29 cells were infected with shRNA, shYAP1#1 and shYAP1#2, in order to abolish YAP1 expression in HT29 cells. Since TAZ protein is not expressed in HT29 cells, shRNA for TAZ was used as control (shCtrl). The infected cells were harvested for RNA and protein at zero and eight days post-confluence. To determine the efficiency of YAP knockdown, the expression of YAP1 and YAP1 target genes including CTGF and CYR61 was first evaluated at the transcript level in newly confluent shYAP1 cells relative to shCtrl cells. *YAP1*, *CYR61* and *CTGF* were all decreased significantly relative to shCtrl in both shYAP1#1 and shYAP1#2 cells (Figure 3A). Repression of YAP1 expression was confirmed with both shYAP1 expressing HT29 cells while TAZ remained below the detection level (Figure 3B).

### 3.4. Effect of YAP1 Knockdown on Intestinal Stem Cell Marker Expression

To evaluate the effect of YAP1 knockdown on stemness, the mRNA levels of the stem cell markers used above were assayed in shYAP1 cells. As shown in Figure 4, a reduction of *LGR5* and *PROM1* was noted in shYAP1 cells compared to shCtrl cells. Western blot analysis confirmed a reduced expression of LGR5 protein in shYAP1 HT29 cells compared with control.

### 3.5. The Effect of YAP1 Knockdown on Intestinal Epithelial Cell Differentiation

To investigate the effect of YAP1 on cell differentiation, qPCR analysis was first performed to detect the effect of YAP knockdown on the differentiation of absorptive, enteroendocrine, goblet and Paneth cell lineages. The results show that the expression of goblet cell markers, mucin 2 (*MUC2*) and trefoil factor-3 (*TFF3*) was increased significantly in YAP1 knockdown HT29 cells compared to shTAZ#1 used as control (Figure 5). In addition, expression of absorptive cell markers including sucrase-isomaltase (*SI*) and dipeptidyl peptidase-4 (*DPPIV*), was increased significantly at the transcriptional level in shYAP1 cells. However, removing YAP1 in HT29 cells had no effect on the expression of chromogranin A (*CHGA*) or defensin 5 (*DEFA5*), which are specific enteroendocrine and Paneth cell markers, respectively (Figure 5).

Western blot analysis confirmed that abolishing YAP1 expression in HT29 cells promotes both goblet and absorptive cell differentiation. As shown in Figure 6, the goblet cell markers MUC2 and TFF3 were consistently found to be increased in shYAP1 cells. Higher expression was observed in newly confluent (0PC) as well as in eight-days post-confluent (8PC) shYAP1 cells relative to shCtrl cells (Figure 6A). Indirect immunofluorescence on these cells revealed that goblet cell differentiation triggered by YAP1 abolition is not uniform in the monolayer but leads to a consistent increase in the number of goblet cells in both newly confluent and eight-days post-confluent cells (Figure 6B–E). Absorptive cell differentiation was further investigated by Western blot and indirect immunofluorescence.

As shown in Figure 7, SI was barely detectable in control HT29 cells at all stages. SI was not detected at significant levels in newly confluent shYAP1 cells but was expressed at a high level in eight-days post-confluent cells (Figure 7A). DPPIV was found to be increased in both newly confluent and post-confluent shYAP1 cells relative to shCtrl cells (Figure 7A). Indirect immunofluorescence staining at sub confluency and eight days post-confluence showed that the number of DPPIV expressing cells was increased in YAP1 knockdown HT29 cells compared with the control (Figure 7B–E). It is noteworthy that, as for goblet cells, only a subset of shYAP1 express DPPIV in the HT29 cell monolayer. Differentiation of goblet- and absorptive-like cells in YAP1 knockdown HT29 cells was confirmed by transmission electron microscopy (Figure 8).

To further support our observations on the role of YAP1 in intestinal epithelial cell differentiation, YAP1 expression was knocked down in Caco-2/15 cells using shYAP1#1. Since Caco-2/15 have the potential to differentiate to only absorptive cells, the effect of the YAP1 knockdown was investigated on the expression of the absorptive cell marker SI in five-days post-confluent cells using qPCR and Western blot analyses. A higher expression of SI mRNA was observed in Caco-2/15 cells expressing shYAP1 compared with control cells expressing shLUC (Figure 9A). A significant increase in SI protein was also observed in shYAP1 expressing cells as compared with controls (Figure 9B). Although not further investigated, these results support the finding that YAP1 exerts an inhibitory influence on intestinal cell differentiation.

### 3.6. Influence of YAP1 on Expression of Intestinal Differentiation-Regulating Transcription Factors

As a first step to investigate the mechanism by which the abolition of YAP1 leads to differentiation, the expression of specific transcription factors involved in goblet and absorptive cell differentiation was studied. *CDX2*, *ATOH1*, *HNF1α* and *KLF4* were analyzed by qPCR. As shown in Figure 9, the expression of *CDX2* and *ATOH1* was increased significantly in both sets of YAP1 knockdown cells compared with the control while the other tested transcription factors were not found to be significantly modulated (Figure 10A). Western blot analysis confirmed the important induction of CDX2 protein expression in shYAP1 expressing cells compared to shCtrl in which CDX2 protein is barely detectable (Figure 10B). However, ATOH1 was found to be below detection levels in HT29 under all conditions and was therefore not further analyzed.

### 3.7. YAP1 Controls Differentiation of Intestinal Absorptive and Goblet Cells through CDX2

To explore the involvement of CDX2 as a possible mediator of the pro-differentiation effect of YAP1 abolition in HT29 cells, the expression of CDX2 was knocked down in shYAP1 HT29 cells using shRNA. The shYAP1 stably expressing ± shCDX2 and shCtrl were harvested at five days post-confluence and mRNA and protein fractions were prepared. We first evaluated the efficiency of CDX2 knockdown by measuring CDX2 at transcriptional and protein levels using qPCR and Western blot analysis, respectively. As shown in Figure 11, the expression of CDX2 was significantly diminished in shYAP1+ shCDX2 expressing cells at both mRNA and protein levels. In fact, Western blot analysis showed that the increase in CDX2 resulting from YAP1 abolition is completely neutralized in shYAP1 expressing shCDX2 (Figure 10B).

Further analysis of these five-days post-confluent cells for the expression of absorptive and goblet cell markers showed a significant reduction of three of these markers at both transcript (Figure 12A) and protein levels (Figure 12B). Indeed, MUC2, TFF3 and SI were consistently repressed at comparable levels than those observed in shCtrl cells (Figure 12B) suggesting that the upregulation of these markers in YAP1 knockdown cells mainly results from CDX2 upregulation. One exception was DPPIV for which the expression was not altered by the abolition of CDX2 expression indicating that YAP1 may regulate other transcription factor(s) as well.

Finally, expression of stem cell markers at five days post-confluence was significantly reduced in shYAP1 expressing cells as compared to shCtrl indicating that YAP1 is still modulating some of the stemness-like features even at confluence (Figure 13A). Interestingly, abolition of CDX2 expression in shYAP1 cells restored the levels of *LGR5* and *PROM1* expression observed in the wild-type HT29 population (Figure 13B).

## 4. Discussion

The present study provides new insights into the role of the Hippo pathway in the regulation of intestinal cell functions by showing that YAP1 can specifically repress differentiation for both absorptive and secretory cell lineages and confirming its involvement in stemness maintenance.

One difficulty in assessing the involvement of the Hippo pathway is the need to consider YAP1 as well as its paralog TAZ since both transcriptional coactivators can combine with other factors to promote the expression of target genes [32]. As shown herein for Caco-2 cells, most colorectal cell lines express both YAP1 and TAZ resulting in limited effects in knockdown approaches unless double repression is performed [59]. In this context, the use of HT29, which only expresses YAP1, is a clear advantage. Another difficulty for differentiation-based studies is that most available intestinal cell models display very limited differentiation potential and no multipotency, HT29 cells being the exception since these cells can give rise to both absorptive and goblet cells, the two main intestinal cell types [1], under certain conditions [42,43,44]. Furthermore, HT29 cells are well characterized for their stem-like cell subpopulation [41].

The endogenous restricted expression of YAP1/TAZ in a few non-Paneth cells located in the lower portion of the crypts that was observed herein in the human intestine confirmed the similarity with its distribution that has been reported in the intestine of animal models in which a functional association with stem cells was demonstrated [34,36]. Thus, using the HT29 cell model, we first assessed the influence of YAP1 on stemness properties by showing a reduction of two intestinal stem cell markers, LGR5 and PROM1, in cells knocked down for YAP1, which is consistent with the fact that YAP1/TAZ support stem cell properties in the intestine [36,40] and are required for stemness maintenance and colony formation in colorectal cancer cells [60,61].

The striking effect of YAP knockdown on HT29 cell differentiation was more surprising. Indeed, YAP1 and TAZ knockdown performed in various colorectal cell lines showed that they can affect proliferation, apoptosis, migration and invasion [59,60], but, as far as we know, no report has addressed intestinal differentiation, although previous in vivo studies have shown that overexpression of YAP1 directly or through elimination of Mst1 and Mst2, which both enhance crypt cell proliferation, was accompanied by a significant impairment of differentiation for all cell types [60,62]. According to our results, YAP1 knockdown in HT29 cells resulted in a sharp increase in the expression of absorptive (SI, DPPIV) and goblet (MUC2, TFF3) cell markers while Paneth (DEFA5) and enteroendocrine (CHGA) cell markers did not change. Upregulation of SI, DPPIV, MUC2 and TFF3 was confirmed at the protein level. As expected from the normal villus epithelium [1], even at eight days post-confluence, only a subset of cells exhibited goblet-like cell properties while a larger proportion of the cells displayed absorptive-like cell properties in immunofluorescence and electron microscopy. YAP1 knockdown in Caco-2/15 cells also resulted in an increase in SI expression, thus confirming the findings in another cell model widely used for studying human intestinal cell differentiation [46,47]. However, because Caco-2/15 cells express high levels of both YAP1 and TAZ and can only differentiate into absorptive cells, further experiments with this cell model were not pursued.

The path by which YAP1 ablation induced absorptive and goblet cell differentiation was then investigated by screening for changes in the expression of the main known regulators of intestinal cell differentiation, namely CDX2 and HNF1α, two master regulators of intestinal epithelial cells [21,22,63,64], ATOH1, a downstream effector of the Notch pathway for promoting secretory lineages [16,18], and KLF4 are required for goblet cell differentiation [26]. The expression of *CDX2* and *ATOH1* was found to be significatively increased in shYAP1 HT29 cell lines but only CDX2 was detected in shYAP1 cells at the protein level. As shown previously, normal HT29 is among the few intestinal cell lines not expressing CDX2 [65]. The involvement of CDX2 in absorptive and goblet cell differentiation in response to YAP1 ablation was then confirmed by observing a return to the basal level of SI, MUC2 and TFF3 expression in shYAP1 + shCDX2 transduced cells while DPPIV remained elevated. These results are consistent with the fact that CDX2 has been found to specifically interact with the promoters of SI [21,66], MUC2 [67,68] and TFF3 [69] to directly stimulate their expression in various intestinal cell models including HT29 and Caco-2 [66,68]. The lack of effect of CDX2 knockdown on DPPIV suggests that other transcriptional factors may be stimulated by the ablation of YAP1. For instance, DPPIV depends on factors such as HNF1α, HNF1β, USF-1 and USF-2 [70,71], which with the exception of HNF1α have not been investigated. Considering the general pro-differentiation effect of YAP1 abolition noted on morphological differentiation, it is likely that other transcription factors involved in the regulation of epithelial cell polarization such as GATA4 [22,72] could be involved. In this context, it is interesting to note that the levels of LGR5 and PROM1 found to be repressed by the ablation of YAP1 were restored in shCDX2 expressing cells, suggesting that this factor may act in the repression of stem cell-related genes in intestinal cells in agreement with previous work [73].

While CDX2 has been shown to regulate YAP1 expression in intestinal cells [74], the mechanism by which YAP1 represses CDX2 remains elusive. As shown previously, a lack of CDX2 expression in HT29 is obviously not the result of a somatic mutation, nor does it appear to result from epigenetic silencing [65,75], although a recent study suggests that minimal expression of CDX2 could be restored in HT29 after treatment with a histone deacetylase 4/5 inhibitor [76]. The antagonist influence of Oct4 on Cdx2 transcription in early embryonic development [77] was thus investigated based on the negative correlation reported between the expression of the two factors in colorectal cancer [78] but no modulation of OCT4 expression was observed in response to YAP1 ablation in HT29 cells. In another study, Shang et al. reported that the factor ASCL2 was overexpressed in a subset of colorectal cell lines and that its knockdown in HT29 cells resulted in approximately a 2× increase in CDX2 expression, through a transcription-dependent mechanism [79]. However, the very low level of ASCL2 in HT29 cells observed herein and the lack of modulation of its expression in response to YAP1 ablation indicates that this mechanism cannot account for the pro-differentiation effects observed herein. It has also to be pointed out that the identity of the HT29 cells used in the study of Shang et al. [79] could be questioned considering that wild type HT29 do not express detectable levels of CDX2 protein as shown herein and elsewhere [65,80,81]. Further work will thus be needed to elucidate the molecular mechanism by which YAP1 represses CDX2 expression in intestinal cells.

## 5. Conclusions

The present study provides new evidence that, in addition to regulating proliferation, survival, migration and invasion as well as stemness, the main downstream effector of the Hippo pathway YAP1 exerts a major effect on the repression of differentiation for absorptive and goblet cells, the two main intestinal cell lineages. The YAP1 effect is mediated in part by the repression of the expression of CDX2, one of the master regulators of intestinal growth and differentiation, but it also likely involves other transcription factors. This finding indicates that the Hippo effector YAP1 not only regulates intestinal stemness [34,36] but also restrains differentiation by repressing expression of pro-differentiation transcription factors such as CDX2. As depicted in Figure 14, this appears to represent another way of repressing intestinal differentiation in the proliferative compartment of the crypt, which adds to other previously identified mechanisms involving polycomb repressive complex 2 and histone deacetylases that mainly inhibit absorptive cell differentiation under a CDX2 independent manner [52,82].

## Figures and Tables

**Figure 1 cells-09-01895-f001:**
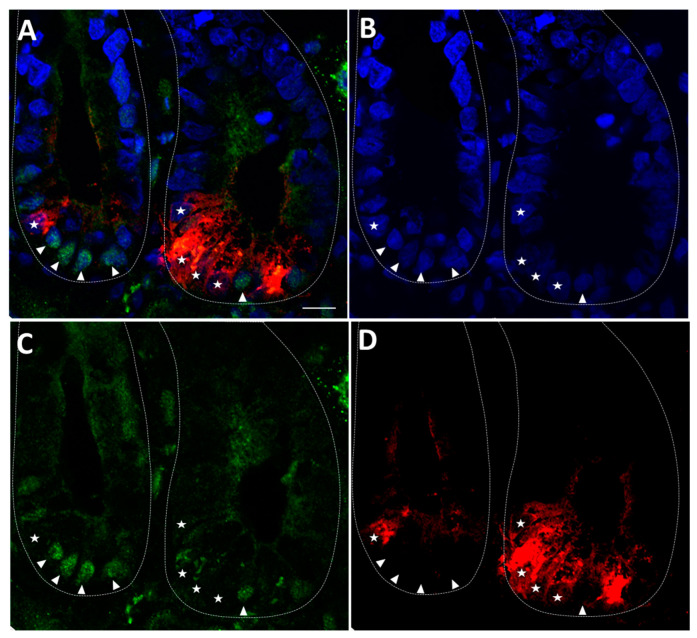
Nuclear expression of YAP1/TAZ in human intestinal crypt cells. Representative confocal imaging for the detection of YAP1/TAZ (green) (**A**,**C**), DEFA5 (red) (**A**,**D**) and DAPI (blue) (**A**,**B**) in the adult small intestine. Nuclear expression of YAP1/TAZ in some of the cells located at the bottom of the crypts was observed (arrowheads) except in Paneth cells (stars). Scale bar is equal to 10 μm.

**Figure 2 cells-09-01895-f002:**
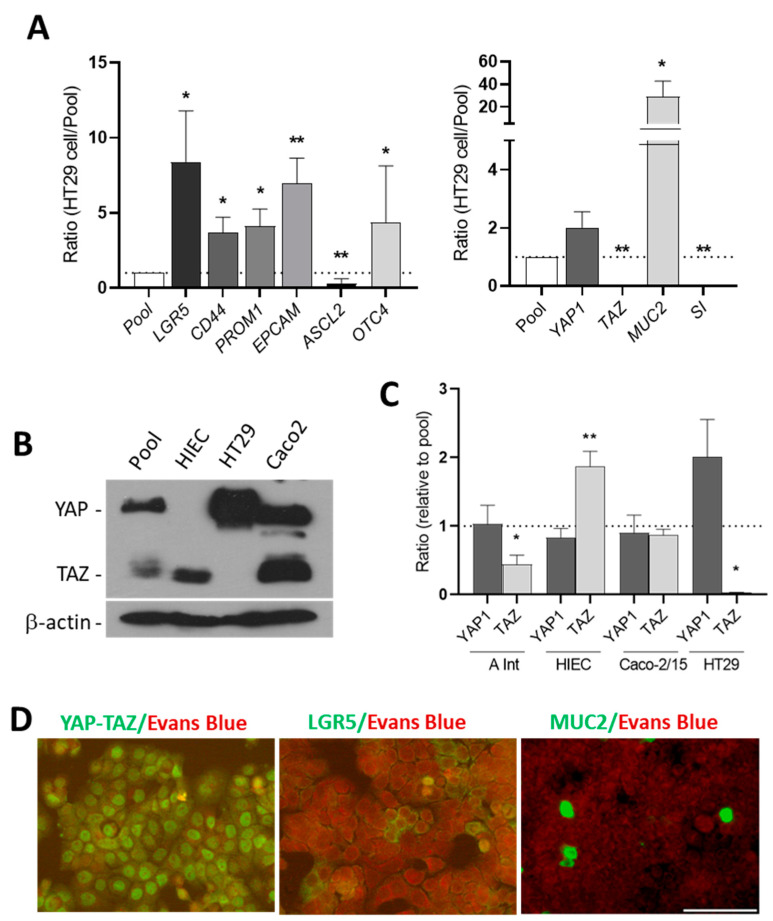
Expression of stem cell markers, Hippo effectors, goblet and absorptive cells markers in HT29 cells. (**A**) Expression of *LGR5*, *CD44*, *PROM1*, *EPCAM*, *YAP1*, *TAZ*, *MUC2* and *SI* transcripts in HT29 cells relative to a pool of cancer cells. * *p* < 0.05, ** *p* < 0.01. (**B**) Western blot analysis showing expression of YAP protein in HT29 cells in which the TAZ protein was consistently below detectable levels. Both YAP and TAZ proteins were found to be expressed by Caco-2 cells while only TAZ was detectable in HIEC. β-actin was used as a loading control. (**C**) The expression of YAP1 and TAZ was also investigated at the transcript levels in the adult small intestine (A Int) and the intestinal cell lines relative to the pool. Statistical significance for YAP1 vs. TAZ (paired T test): * *p* < 0.05, ** *p* < 0.005, *n* ≥ 3. (**D**) Indirect immunofluorescence of HT29 cells confirmed the presence of the YAP protein in a large proportion of the cells while a few LGR5 and MUC2 positive cells were detected in the normal HT29 cells. Scale bar = 50 μm.

**Figure 3 cells-09-01895-f003:**
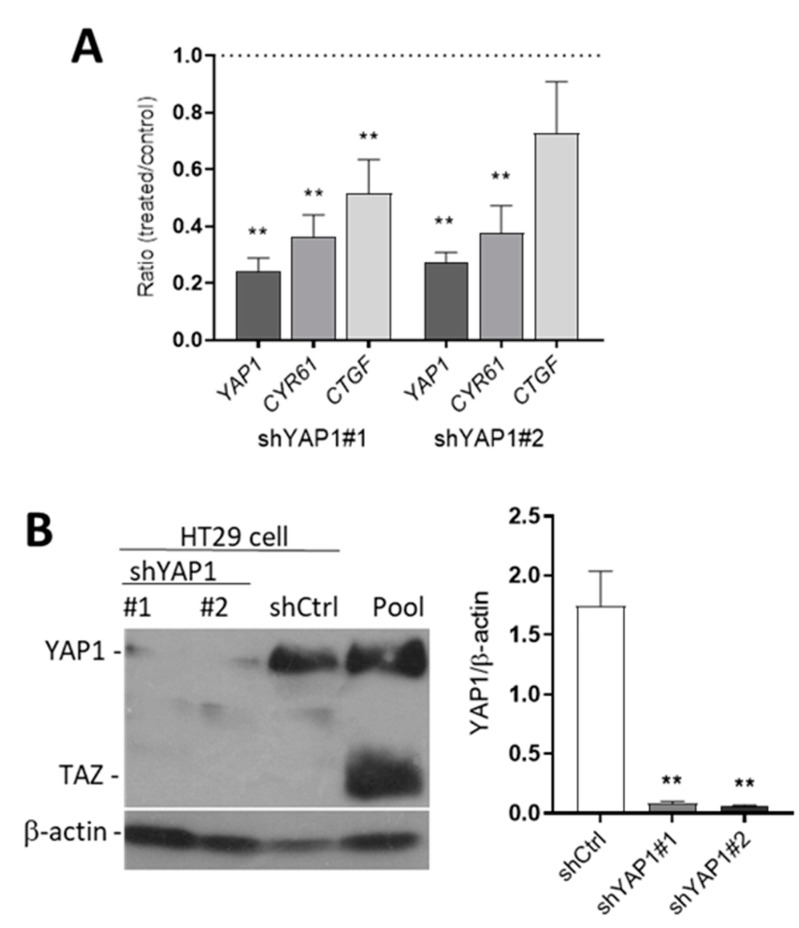
Efficiency of YAP1 knockdown in HT29 cells. (**A**) qPCR analyses were performed to detect the expression of *YAP1* and two of its target genes *CYR61* and *CTGF* in shYAP1 stably expressing cells using two specific sequences (identified as shYAP1 #1 and #2) relative to shCtrl. (**B**) Western blot analysis showing the expression of YAP1 protein in shYAP1#1, shYAP1#2 and shCtrl expressing cells. A cancer cell pool was used as reference for YAP1 and TAZ detection. β-actin was used as a loading control. ** *p* < 0.005.

**Figure 4 cells-09-01895-f004:**
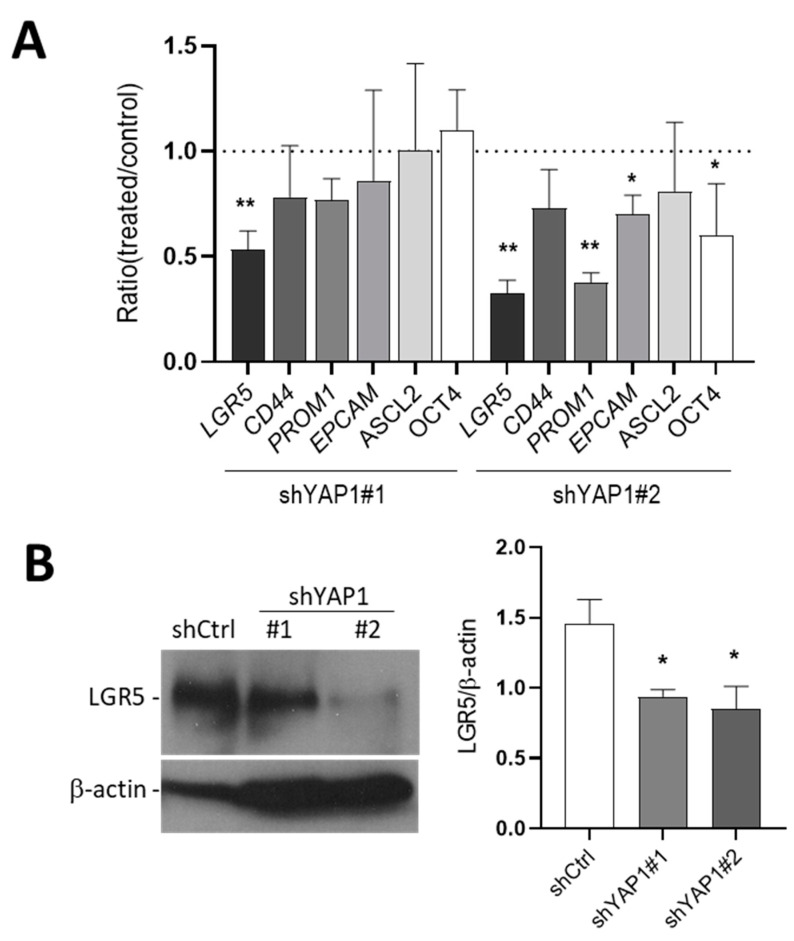
Effect of YAP1 knockdown on stem cell marker expression in subconfluent HT29 cells. (**A**) Transcript expression of the stem cell markers *LGR5*, *CD44*, *PROM1*, *EPCAM*, *ASCL2* and *OCT4* in YAP1 knockdown cells relative to shCtrl (dotted line). (**B**) Reduction in LGR5 protein in both shYAP1 cell lines compared with shCtrl. β-actin was used as a loading control. * *p* < 0.05, ** *p* < 0.005.

**Figure 5 cells-09-01895-f005:**
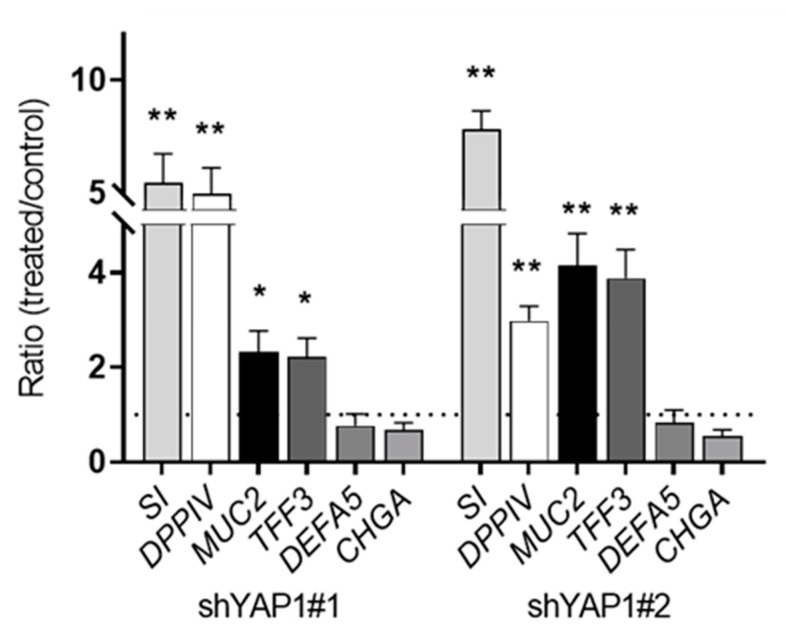
Expression of intestinal cell lineage markers in response to YAP1 knockdown. qPCR analyses of absorptive cell markers *SI* and *DPPIV*, goblet cell markers *MUC2* and *TFF3*, Paneth cell marker *DEFA5* and enteroendocrine cell marker *CHGA* were analyzed in shYAP1#1 and shYAP1#2 expressing cells relative to shCtrl cells (dotted line). * *p* < 0.05, ** *p* < 0.005.

**Figure 6 cells-09-01895-f006:**
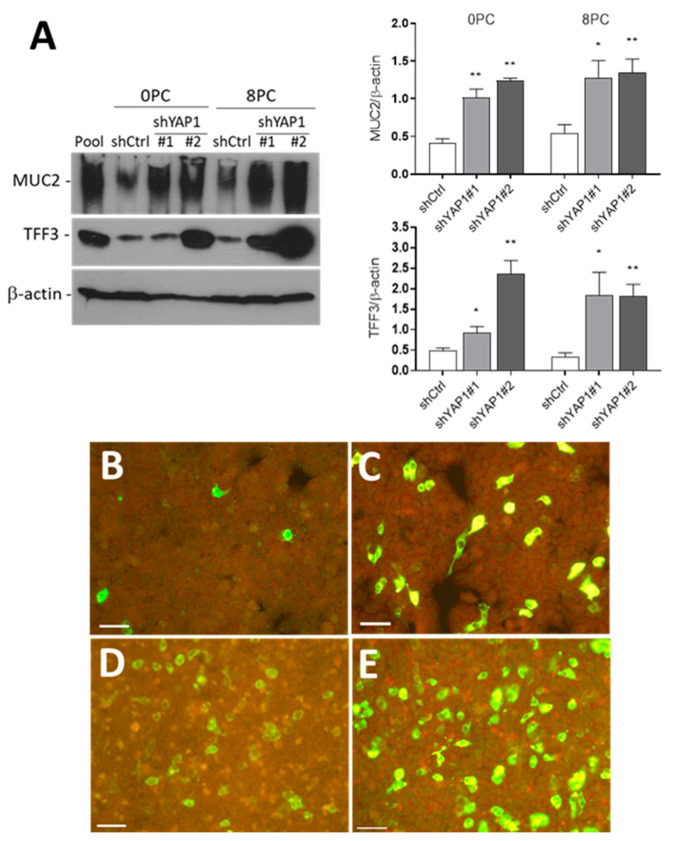
YAP1 knockdown stimulates goblet cell differentiation. (**A**) Representative Western blot analysis and data compilation from three separate experiments showing higher expression of MUC2 and TFF3 in shYAP1#1 and shYAP1#2 expressing cells relative to shCtrl cells at both Day 0 (0PC) and Day 8 (8PC) post-confluence. β-actin was used as a loading control. * *p* < 0.05, ** *p* < 0.005. (**B**–**E**) Indirect immunofluorescence analysis for the detection of MUC2 positive cells in shCtrl (**B**,**D**) and shYAP1#2 (**C**,**E**) at 0PC (**B**,**C**) and 8PC (**C**,**E**). Bar = 50 µm.

**Figure 7 cells-09-01895-f007:**
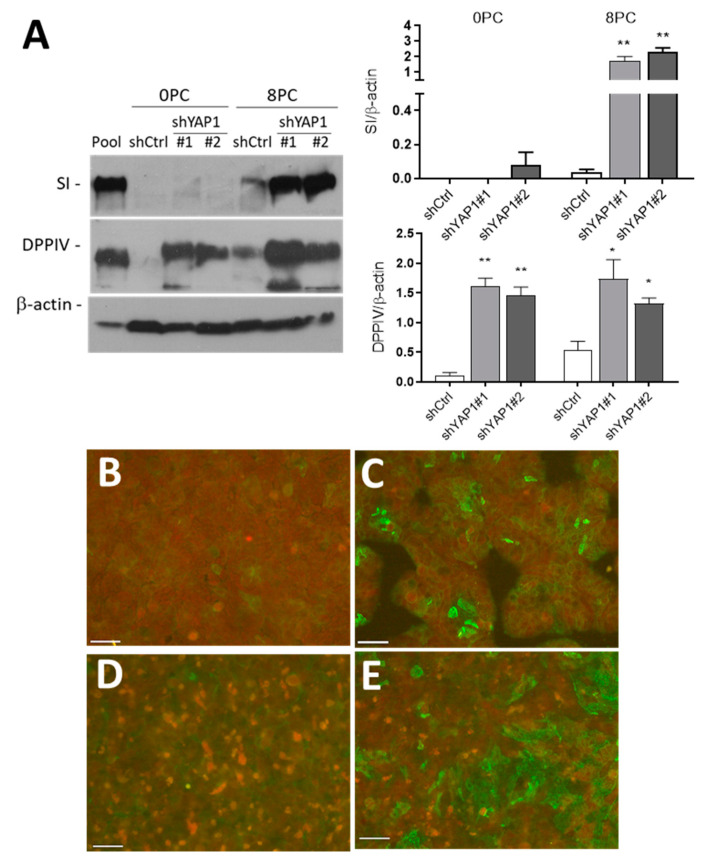
YAP1 knockdown stimulates absorptive cell differentiation. (**A**) Representative Western blot analysis and data compilation from three separate experiments showing higher expression of SI and DPPIV in shYAP1#1 and shYAP1#2 expressing cells relative to shCtrl cells at both Day 0 (0PC) and Day 8 (8PC) post-confluence. β-actin was used as a loading control. * *p* < 0.05, ** *p* < 0.005. (**B**–**E**) Indirect immunofluorescence analysis for the detection of DPPIV positive cells in shCtrl (**B**,**D**) and shYAP1#2 (**C**,**E**) at 0PC (**B**,**C**) and 8 PC (**C**,**E**). Bar = 50 µm.

**Figure 8 cells-09-01895-f008:**
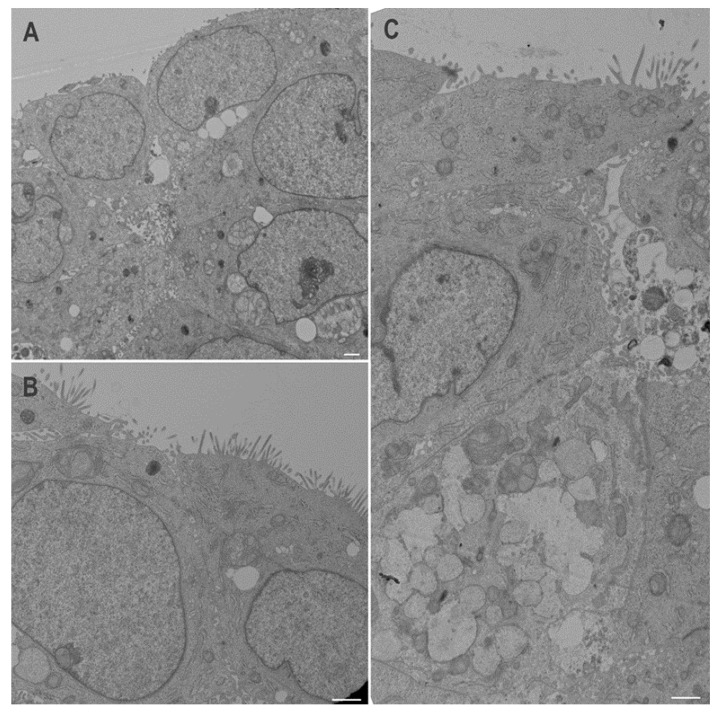
Transmission electron microscopy of HT29 stably expressing shCtrl or shYAP1 at 8PC. (**A**) Control cells expressing shTAZ are small and multilayered and express limited differentiation characteristics. (**B**,**C**) Cells expressing shYAP1 are larger and exhibit some absorptive-like features such as regular microvilli (**B**) and goblet cell-like features such as mucinous granule forming goblet-like aggregates (lower part of (**C**)). Scale bars = 1 μm.

**Figure 9 cells-09-01895-f009:**
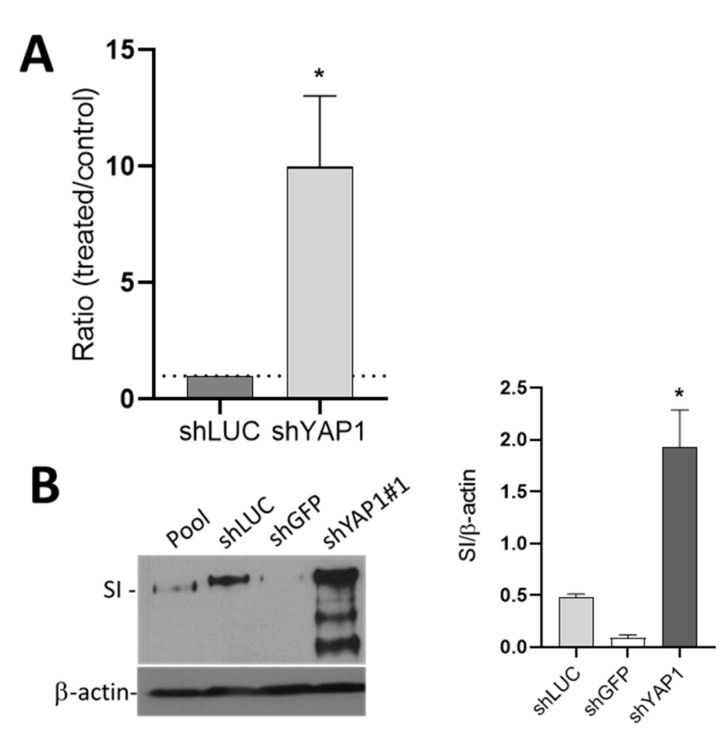
The effect of shYAP1 expression on Caco-2/15 cell differentiation. (**A**) Transcript expression of the absorptive cell marker SI in Caco-2/15 cells expressing shYAP1 relative to shCtrl (shLUC). (**B**) Western blot analysis and data compilation showing higher expression of SI in shYAP1 Caco-2/15 cells relative to shCtrl at five days post-confluence. β–actin was used as a loading control. Statistical comparison between shYAP1 vs. shLUC: * *p* < 0.05.

**Figure 10 cells-09-01895-f010:**
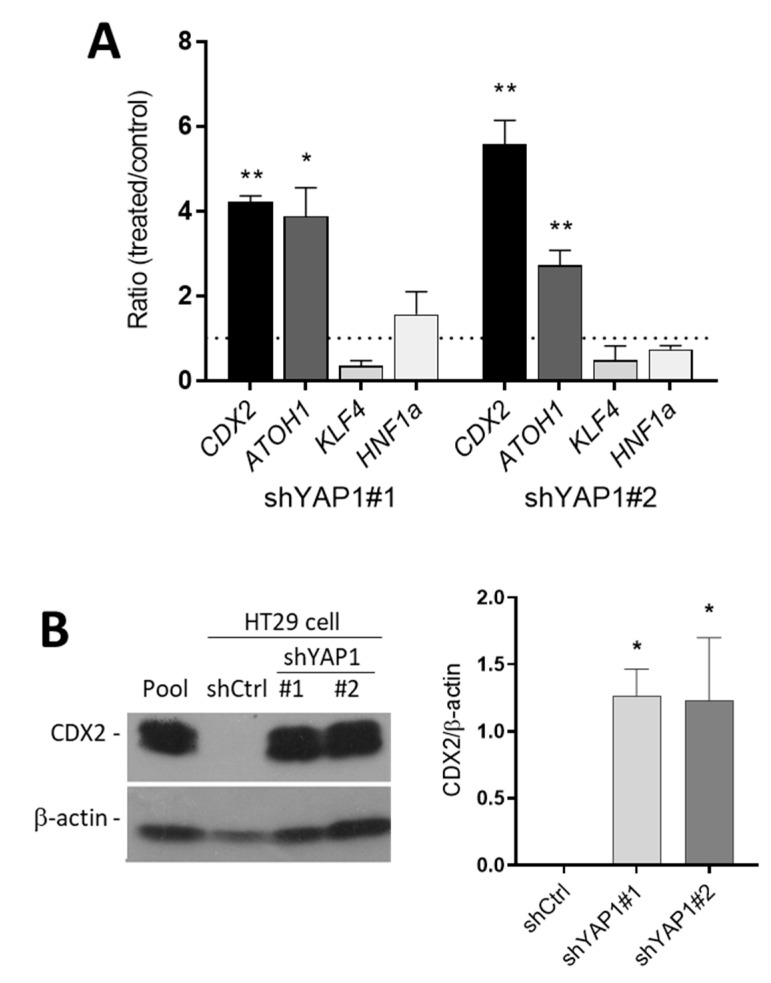
Expression of intestinal differentiation-regulating transcription factors in response to YAP1 knockdown. (**A**) qPCR analysis of the expression of various transcription factors involved in intestinal epithelial cell differentiation in shYAP1#1 and shYAP1#2 expressing cells relative to shCtrl (dotted line) showing the significant increase in expression of *CDX2* and *ATOH1* in YAP1 knockdown cells. (**B**) Representative Western blot analysis and data compilation from three separate experiments showing higher expression of CDX2 protein in YAP1 knockdown HT29 cells. β-actin was used as a loading control. * *p* < 0.05, ** *p* < 0.005.

**Figure 11 cells-09-01895-f011:**
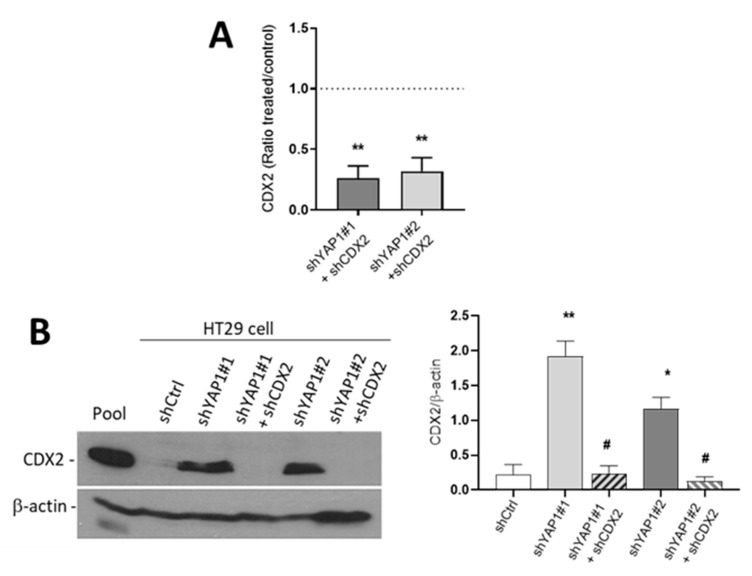
Efficiency of CDX2 knockdown in shYAP1 expressing HT29 cells. (**A**) qPCR analyses were performed to evaluate the expression of *CDX2* in shCDX2 stably expressing shYAP1#1 and -#2 cells relative to normal shYAP1#1 and -#2 cells. (**B**) Representative Western blot analysis showing the expression of CDX2 protein in shCtrl vs. shYAP1#1 and shYAP1#2 ± shCDX2 expressing cells and data compilation of three separate experiments. A cancer cell pool was used as reference for CDX2 detection. β-actin was used as a loading control. * significant vs. shCtrl; # significant vs. shYAP1; */# *p* < 0.05, ** *p* < 0.005.

**Figure 12 cells-09-01895-f012:**
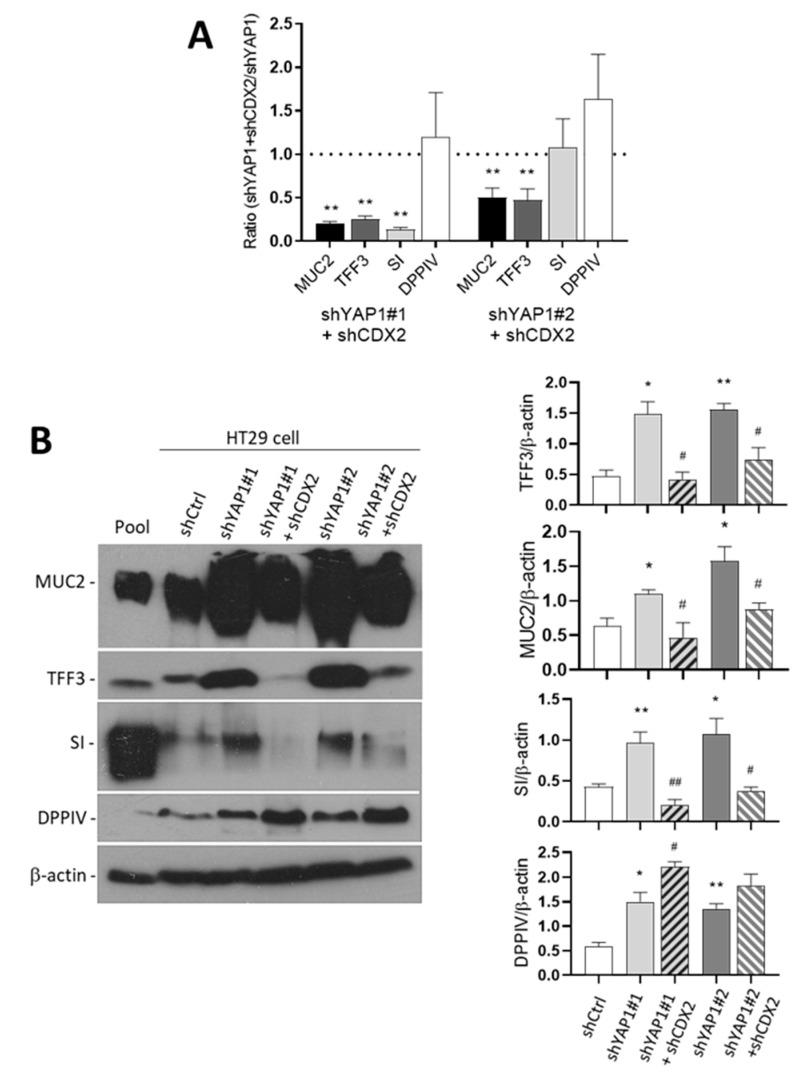
CDX2 mediates the upregulation of most of the intestinal differentiation markers in shYAP1 cells. (**A**) qPCR analysis showing that the abolition of CDX2 in shYAP1 cells resulted in a significant reduction of *MUC2*, *TFF3* and *SI* expression while it remained without effect on *DPPIV*. (**B**) Representative Western blot analysis showing the expression of MUC2, TFF3, SI and DPPIV in shCtrl vs. shYAP1#1 and shYAP1#2 ± shCDX2 expressing cells and data compilation of three separate experiments. At the transcript level, the upregulation of these markers in shYAP1 cells appears to depend upon CDX2 since its abolition results in the restoration of HT29 basal levels for MUC2, TFF3 and SI while DPPIV appears to remain unaffected. A pool was used as a positive control. β-actin was used as a loading control. * significant vs. shCtrl; # significant vs. shYAP1; */# *p* < 0.05, **/## *p* < 0.005.

**Figure 13 cells-09-01895-f013:**
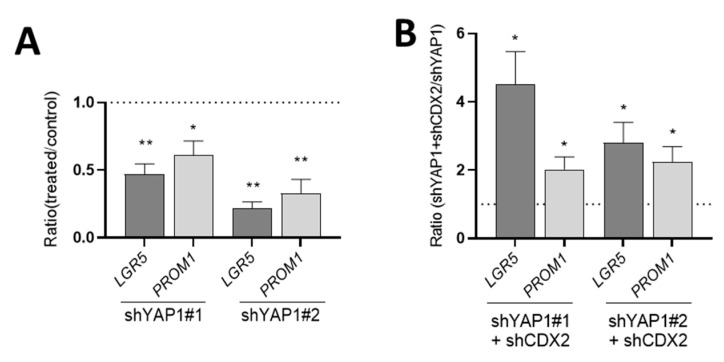
Abolition of CDX2 expression restores control LGR5 and PROM1 levels in five-days post-confluent shYAP1 cells. (**A**) qPCR analysis showing that the abolition of YAP1 resulted in a significant reduction of *LGR5* and *PROM1* expression. (**B**) qPCR analysis showing that the abolition of CDX2 in shYAP1 cells resulted in a significant increase in *LGR5* and *PROM1* expression relative to shYAP1 control cells. A: * significant vs. shCtrl, B: * significant vs. shYAP1; * *p* < 0.05, ** *p* < 0.005.

**Figure 14 cells-09-01895-f014:**
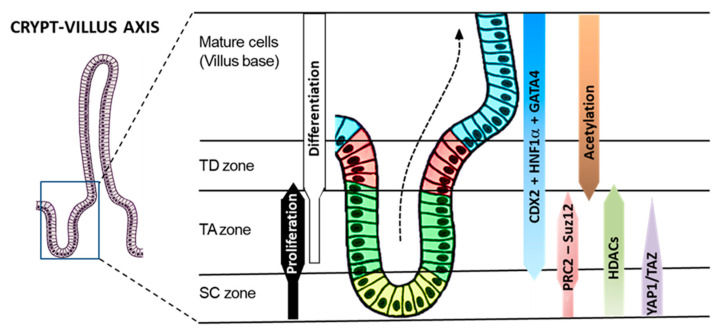
Integration of the molecular mechanisms regulating intestinal differentiation. All steps of intestinal epithelial cell differentiation occur in the crypt so that only fully mature cells reach the villus [1,8]. The crypt is divided into three distinct compartments: the stem cell (SC), the transit amplifying (TA) and terminal differentiation (TD) zones. Intestinal cell differentiation relies on pro-differentiation factors such as CDX2, HNF1α and GATA4 which are expressed in the TA and TD zones. Goblet cells differentiate in the TA zone, but absorptive cell differentiation is restrained by epigenetic mechanisms involving polycomb repressive complex 2 (PRC2-Suz12) and histone deacetylases (HDACs), which allow proliferation in the TA zone, ensuring a larger proportion of absorptive cells in the TD zone. YAP1/TAZ appear to act at an earlier phase, which involves stemness maintenance and inhibition of both absorptive and secretory lineages through a repression of pro-differentiation transcription factors such as CDX2. Adapted from [1,8].

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
