# Peer review of "The Hippo Pathway Effector YAP1 Regulates Intestinal Epithelial Cell Differentiation"

_cells, 2020, doi:10.3390/cells9081895_

Round 1

Reviewer 1 Report

The authors demonstrated YAP1/TAZ in human intestinal crypt cells and that Paneth cells did not express them (Figure 1). Thereafter they used HT29 cells all through the experiments. First, they confirmed that HT29 cells expressed stem cell markers including LGR5, CD44, PROM1, EPCAM, and OCT4, and the goblet cell marker, MUC2, but did not the absorptive cell maker, SI, or TAZ (Figure 2). YAP1 silencing suppressed the expression of TEAD target genes and stem cell markers such as LGR5 (Figure 3 and Figure 4). On the contrary, YAP1 silencing enhanced the goblet cell markers and the absorptive cell markers but not the markers of enteroendocrine and Paneth cells (Figure 5). Immunofluorescence revealed that YAP1 silencing-promoted differentiation of goblet and absorptive cells was not homogeneous in HT29 cells (Figure 6 to Figure 8). YAP1 silencing increased the expression of CDX2 and ATOH1 mRNAs, but as ATOH1 protein was not detected, the authors focused on CDX2 (Figure 9). The additional silencing of CDX2 abolished the effect of YAP1 silencing on MUC2, TFF3, and SI, but not on DPPIV (Figure 11). Moreover, CDX2 silencing recovered the expression of stem cell markers in YAP1-depleted HT29 cells. Based on these findings, the authors concluded “YAP1 exerts a major effect on the repression of differentiation for absorptive and goblet cells.”

The experiments were logically designed and performed and the story is simple and straightforward. However, I have the following comments.

  • I am not familiar with HT29 cells. But these cells are colon cancer cells. Even though they can differentiate into goblet cell-like and absorptive cell-like cells, HT29 cells are not intestinal stem cells. The authors should reconsider the title and make clear the limitation of their study. The title “The Hippo Pathway Effector YAP1 Regulates Human Colon Cancer HT29 Cell Differentiation” will be more appropriate.
  • To complete logic, the authors should show that enforced CDX2 expression enhances the expression of MUC2, TFF3, and SI in HT29 cells.
  • What is the molecular mechanism by which YAP1 represses CDX2? This is the question that most readers will raise after reading this paper. The authors should discuss this point.

Author Response

We thank the reviewers for their positive comments and their suggestions for improving the presentation.

Reviewer 1

Specific comments:

  1. I am not familiar with HT29 cells. But these cells are colon cancer cells. Even though they can differentiate into goblet cell-like and absorptive cell-like cells, HT29 cells are not intestinal stem cells. The authors should reconsider the title and make clear the limitation of their study. The title “The Hippo Pathway Effector YAP1 Regulates Human Colon Cancer HT29 Cell Differentiation” will be more appropriate

RESPONSE:  We agree that HT29 are not stem cells. As now more clearly specified in the last paragraph of the introduction, they have been characterized to contain a certain proportion of stem-like cells (ref #41) as other colon cancer cells but were used in the present study as a very well characterized model for studying human intestinal epithelial differentiation. More information about the characteristics and usefulness of HT29 as an intestinal cell model have been provided in the text (Material and Methods, section 2.1).

Also, in response to a suggestion from the other reviewer, we have included some data about the effect of the expression of shYAP1 in Caco-2/15 cells, another colon cell line which can differentiate into absorptive cells. As detailed in the M&M section 2.1, HT29 and Caco-2 cells are the only two intestinal cell models that have the ability to differentiate into small intestinal epithelial cells. With this addition, we think that the title remains adequate. Nevertheless, we have added the sentence "The utility and limitations of these models (referring to HT29 and Caco-2) are well documented (ref 42-44, 46-48)" in the M&M, section 2.1.

  1. To complete logic, the authors should show that enforced CDX2 expression enhances the expression of MUC2, TFF3, and SI in HT29 cells.

RESPONSE: The enforcement of CDX2 expression on the expression of MUC2, TFF3 and SI has been well-documented in the past using intestinal cell models including HT29 and Caco-2 cells. More information and specific references have been provided in the Discussion (second to last paragraph). 

  1. What is the molecular mechanism by which YAP1 represses CDX2? This is the question that most readers will raise after reading this paper. The authors should discuss this point.

RESPONSE: The mechanism by which YAP1 may control CDX2 expression has been specifically addressed in the last paragraph of the discussion. While epigenetic mechanisms do not appear to play a significant role on CDX2 expression in intestinal cell models, the potential role of specific factors such as OCT4 and ASCL2 has been addressed and does not seem to be involved either. Therefore, we have added the following sentence at the end of the discussion: " Further work will thus be needed to elucidate the molecular mechanism by which YAP1 represses CDX2 expression in intestinal cells."

Reviewer 2 Report

In this manuscript, Fallah et al. examined Yap function in HT29 colorectal cancer cells. HT29 cells were found to express high levels of Yap and no expression of TAZ. These cells were also chosen for their capacity to differentiate into both absorptive and goblet cell. The authors showed that siRNA knockdown of YAP caused a reduction in stem cell markers and increase differentiation markers including MUC2, DEFA5, and TFF3. In attempt to identify the mechanism underlying the differentiation effects resulting from YAP depletion, the authors knocked down CDX2 in combination with YAP. Knockdown of CDX2 reversed the prodifferentiation effects of YAP knockdown in HT29 cells.

Overall the manuscript is technically sound and sheds insight into the poorly defined role of YAP in normal human intestinal stem cell proliferation and differentiation. Indeed, most studies to date have focused on studying Yap function using murine genetic models but it is unclear whether these models are relevant to Yap’s role in human ISCs. This study therefore provides a new perspective on this important transcription factor. Although this manuscript is suitable for publication in Cells, I would recommend the authors address my concerns below regarding the expression analysis of YAP in human intestinal sections and the reliance on a single intestinal cell line.  

Specific points

1) The image quality of the YAP immunofluoresence in normal human ileum should be improved. The image seems to have extensive background. The DAPI and DEFA5 channels should be presented separately. It is difficult to confirm the absence of YAP expression in Paneth cells. Given that the green staining is present throughout the section, secondary antibody alone controls should be performed. Which commercial antibody was used to stain YAP/TAZ? Excellent Yap specific antibodies are available commercially (e.g. CST - YAP (D8H1X)). These should be included in the analysis. In mouse tissues, under normal conditions, TAZ is weakly expressed. Is this the case in human cells? The authors should complement this analysis with western blots from human intestinal epithelium. Given that functional studies are done in CRC cell line HT29, the authors should also stain human colon.

2) The other CRC cell lines examined express high levels of YAP and TAZ. It would be interesting to knockdown either one of these factors or both in combination to confirm their findings. Alternatively, a pharmacological approach could be tried using Verteporfin.

Author Response

We thank the reviewers for their positive comments and their suggestions for improving the presentation.

Reviewer 2

1) The image quality of the YAP immunofluoresence in normal human ileum should be improved. The image seems to have extensive background. The DAPI and DEFA5 channels should be presented separately. It is difficult to confirm the absence of YAP expression in Paneth cells. Given that the green staining is present throughout the section, secondary antibody alone controls should be performed. Which commercial antibody was used to stain YAP/TAZ? Excellent Yap specific antibodies are available commercially (e.g. CST - YAP (D8H1X)). These should be included in the analysis. In mouse tissues, under normal conditions, TAZ is weakly expressed. Is this the case in human cells? The authors should complement this analysis with western blots from human intestinal epithelium. Given that functional studies are done in CRC cell line HT29, the authors should also stain human colon.

RESPONSE:

As suggested, we have replaced the immunolocalization image for YAP1/TAZ in the small intestinal crypt (Figure 1). As shown in supplemental Figure 1 with conventional indirect immunofluorescence (former immunofluorescence image), the background of the staining in controls (not included at the time of the original submission) is very low. While a predominant YAP/TAZ staining is observed in some nuclei, a low level of cytoplasmic staining is to be expected in some cells since YAP1 and TAZ are subject to cytoplasmic localization and degradation in Hippo active cells. Nevertheless, to ensure a sharper staining, we used confocal microscopy to finalize the acquisitions. As suggested, we have included all channels separately and as a merge clearly showing that in the lower crypt, positive nuclei for YAP1/TAZ staining occurs only in non-Paneth cells.    

The antibody used was the D24E4 rabbit monoclonal anti YAP/TAZ obtained from Cell Signaling that works well in both western blot (detecting both YAP1 and TAZ) and immunofluorescence. We have noted the Yap specific antibody suggested but it was not possible to test it within the 10 days allowed for the revision.

As shown in the new Figure 2C, the relative expression of YAP1 and TAZ was evaluated by qPCR in samples of human small intestine. Based on the relative correlation between transcripts and proteins for YAP1 and TAZ expression in the 3 cell lines (YAP < TAZ in HIEC, YAP = TAZ in Caco-2, YAP > TAZ in HT29), the YAP1 > TAZ pattern of expression observed in the small intestinal mucosa suggests that lower amounts of TAZ relatively to YAP1 are expressed in the human intestine as reported in the mouse.

While HT29 and Caco-2 are derived from the colon, it is noteworthy that they differentiate into small intestinal epithelial cells (e.g. SI expression), a resurgence of the fetal phenotype (the fetal colon expresses small intestinal markers until 30 weeks of gestation). Thus, we always use samples of small intestine as the normal reference in these studies. 

2) The other CRC cell lines examined express high levels of YAP and TAZ. It would be interesting to knockdown either one of these factors or both in combination to confirm their findings. Alternatively, a pharmacological approach could be tried using Verteporfin.

RESPONSE:

These possibilities were addressed in the initial phases of the project. Of course, since Verteporfin (VP) was easier to use than shRNAs, we tested it. Unfortunately, HT29 and Caco-2 cells were strongly affected by VP used under optimal working concentrations (e.g. 5-10 µM). They only supported 1 µM for up to 5 days but this concentration did not result in any reduction of YAP1 or TAZ levels. For shRNAs, we ran preliminary tests with Caco-2/15 cells. These results have been included in the new Figure 9. As mentioned in the text, Caco-2/15 cells expressing shYAP1 had more SI at both the transcript and protein levels (lines 408-416) thus confirming the inhibitory effect of YAP1 on intestinal cell differentiation (lines 559-561) but as mentioned, we did not continue with this model because of the dual expression of YAP1 and TAZ and the fact that these cells can only differentiate into the absorptive lineage (lines 561-563) preferring to focus on HT29 which express only YAP1 but differentiate into both absorptive and secretory lineages.    

Round 2

Reviewer 1 Report

I still think that the current title is misleading, As the authors have mentioned, previous studies reported that HT29 and Caco-2 cells have stem cell-like properties. Even so the authors of such papers did not use the titles, for instance, such as “Regulatory sequences on the human villin gene trigger the expression of a reporter gene in a differentiating intestinal epithelia cells” or “Clonal analysis of sucrase-isomaltase expression in the human intestinal epithelial cells”. But that is OK. I am not so much particular about this point.

Reviewer 2 Report

The authors have satisfactorily addressed my comments. The manuscript is now suitable for publication in Cells